# Fair Feature Importance Scores for Interpreting Tree-Based Methods and Surrogates

## Abstract

Across various sectors such as healthcare, criminal justice, national security, finance, and technology, large-scale machine learning (ML) and artificial intelligence (AI) systems are being deployed to make critical data-driven decisions. Many have asked if we can and should trust these ML systems to be making these decisions. Two critical components are prerequisites for trust in ML systems: interpretability, or the ability to understand why the ML system makes the decisions it does, and fairness, which ensures that ML systems do not exhibit bias against certain individuals or groups. Both interpretability and fairness are important and have separately received abundant attention in the ML literature, but so far, there have been very few methods developed to directly interpret models with regard to their fairness. In this paper, we focus on arguably the most popular type of ML interpretation: feature importance scores. Inspired by the use of decision trees in knowledge distillation, we propose to leverage trees as interpretable surrogates for complex black-box ML models. Specifically, we develop a novel fair feature importance score for trees that can be used to interpret how each feature contributes to fairness or bias in trees, tree-based ensembles, or tree-based surrogates of any complex ML system. Like the popular mean decrease in impurity for trees, our *Fair Feature Importance Score* is defined based on the mean decrease (or increase) in group bias. Through simulations as well as real examples on benchmark fairness datasets, we demonstrate that our Fair Feature Importance Score offers valid interpretations for both tree-based ensembles and tree-based surrogates of other ML systems.

## 1 Introduction

The adoption of machine learning models in high-stakes decision-making has witnessed a remarkable surge in recent years. Employing these models to assist in human decision processes offers significant advantages, such as managing vast datasets and uncovering subtle trends and patterns. However, it has become increasingly evident that the utilization of these models can lead to biased outcomes. Even when users can discern bias within the model's results, they frequently encounter substantial hurdles when attempting to rectify this bias, primarily due to their inability to comprehend the inner workings of the model and the factors contributing to its bias. When machine learning models impact high-stakes decisions, trust is paramount (Toreini et al., 2020; Rasheed et al., 2022; Broderick et al., 2023). Users, stakeholders, and the general public need to have confidence in the fairness and interpretability of these models. Without comprehensible explanations and the ability to audit model decisions, trust can degrade rapidly.

An incident with the Apple Credit Card in 2019 is a prime example of this. The wife of a long-time married couple applied for an increased credit limit for her card (Vigdor, 2019). Despite having a better credit score and other positive factors in her favor, her application for an increased line of credit was denied. The husband, who had filed taxes together with his wife for years, wondered why he deserved a credit limit 20 times that of his wife. When the couple inquired as to why the credit limit was so different, no one was able to explain the decision to the couple, which created consternation amongst these and other clients on social media who also demanded explanations (Knight, 2019). This led to an investigation by the New York State Department of Financial Services. While the investigation showed that the card did not discriminate based on gender (Campbell, 2021), the inability to provide an interpretation or explanation about the fairness of the algorithm used to

determine credit limits created significant mistrust. Moving forward, it is critical that we have ways of interpreting ML systems based not only on the accuracy of predictions, but also on the fairness of the predictions. As a particular example, we have many ways to interpret how features affect a model's predictions through feature importance scores (Du et al., 2019; Murdoch et al., 2019). Yet, we have no current way of understanding how a feature affects the fairness of the model's predictions. The goal of this paper is to fill in this critical gap by developing a simple and interpretable fair feature importance score.

Countless works have proposed methods to improve fairness in existing models (Zemel et al., 2013; Calmon et al., 2017; Agarwal et al., 2018; Zhang et al., 2018; Lohia et al., 2019; Caton & Haas, 2020), but few have focused on how to interpret models with regards to fairness. We adopt a simple approach and consider interpreting features in decision trees. Why trees? First, trees have a popular and easy-to-compute intrinsic feature importance score known as mean decrease in impurity (MDI) (Breiman, 1973). Second, tree-based ensembles like random forests and boosting are widely used machine learning models, especially for tabular data. Finally, decision trees have been proposed for knowledge distillation of deep learning systems and other black-box systems (Hinton et al., 2015; Gou et al., 2021). Decision trees have also more recently been proposed for use as interpretability surrogates for deep learning systems (Guidotti et al., 2018; Schaaf et al., 2019).

In this work, we develop a straightforward and intuitive metric for calculating fair feature importance scores in decision trees. Our Fair Feature Importance Score (*FairFIS*) reveals which features lead to improvements in the fairness of a model's predictions and which degrade fairness or contribute to the model's bias. Additionally, we show how *FairFIS* can be used to explain the fairness of predictions in tree-based ensembles and through tree-based surrogates of other complex ML systems.

## 1.1 RELATED WORKS

To promote trust, transparency, and accountability, there has been a surge in recent research in interpretable ML; see reviews of this literature by Molnar (2020); Lipton (2018) for more details. Interpretable ML (or explainable AI) seeks to provide human understandable insights into the data, the model or a model's output and decisions (Allen et al., 2023; Murdoch et al., 2019). One of the most popular interpretations is feature importance, which measures how each feature contributes to a model's predictions. There are a wide variety of model-specific feature importance measures like the popular mean decrease in impurity (MDI) for decision trees (Louppe et al., 2013) or layer-wise relevance propagation (LRP) for deep learning (Samek et al., 2021), among many others. Several proposed model agnostic measures of feature importance include Shapley values, feature permutations, and feature occlusions (Mase et al., 2021; Chen et al., 2018).

Another notable category of interpretability-enhancing techniques involves surrogate models. A surrogate model is a simplified and more interpretable representation of a complex, often black-box model (Samek & Müller, 2019). Surrogate models are designed to approximate the behavior of the original model while being easier to understand, faster to compute, or more suitable for specific tasks such as optimization, sensitivity analysis, or interpretability; examples include linear models, decision trees or Gaussian processes. One of the most well-known surrogates for interpretability is LIME (Local Interpretable Model-Agnostic Explanations) (Ribeiro et al., 2016); this approach builds a simple and interpretable (usually linear) model to interpret a local sub-region of the input space. Global surrogates, on the other hand, build a second surrogate model to approximate the global behavior and all the predictions of the original model. Decision trees have been proposed as potential global surrogates as they are fast, simple and interpretable, and as a fully grown decision tree can exactly reproduce the predictions of the original model on the training data (Blanco-Justicia & Domingo-Ferrer, 2019). On a related note, decision trees have played a crucial role in an associated field known as knowledge distillation, where simplified surrogates of complex models are crafted to mimic the complex model's predictions (Hinton et al., 2015; Gou et al., 2021). Although knowledge distillation focuses on prediction, it is worth noting that if predictions from surrogate decision trees prove to be accurate, they can also be harnessed for interpretation (Yang et al., 2018; Sagi & Rokach, 2021; Wan et al., 2020).

Separate from interpretability, fairness is another critical component to promote trust in ML systems. There has been a surge of recent literature on fairness (Chouldechova & Roth, 2018; Friedler et al., 2019). And while many methods have been developed to mitigate bias in ML systems (Zhang

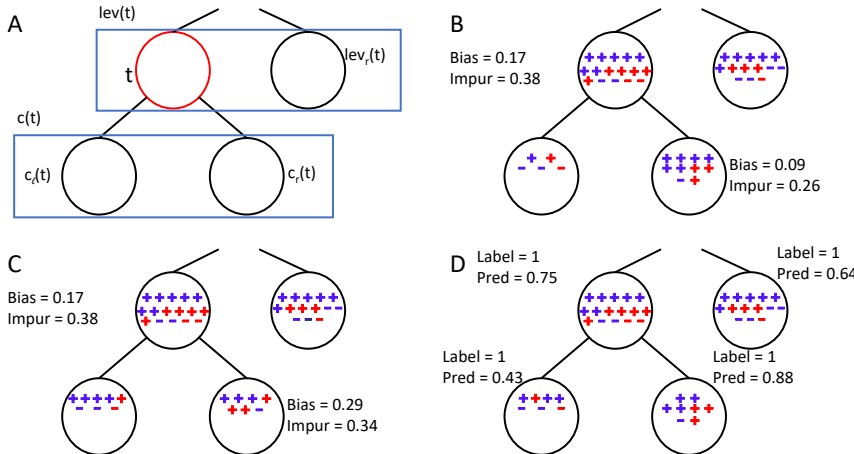

Figure 1: Schematic trees to illustrate *FIS* and *FairFIS*. Panel A illustrates the level of node $t$ and the child level of $t$ that are used to calculate *FairFIS*. Panels B-D illustrate classification trees with pluses and minuses denoting positive and negative labels respectively and red and blue denoting the majority and minority groups respectively. Panels B and C show the *Bias* and weighted impurity (Gini Index) at node or level $t$ and that of the children of node $t$. In Panel B, notice the *Bias* decreases between the parent and child level, resulting in a positive *FairFIS* for that split. Differently, in Panel C, the *Bias* increases, resulting in a negative *FairFIS* for that split. Panel D illustrates why we must use soft predictions versus hard labels when computing *FairFIS*.

et al., 2018; Grari et al., 2019; Agarwal et al., 2018), very few of these papers have additionally focused on interpretability. Yet, many have called for improving interpretability in the context of fairness Agarwal (2021); Wang et al. (2023). Notably, there are a few recent examples that seek to address this. Begley et al. (2020) introduces a new value function that measures fairness for use within Shapley values; although this is an interesting and relevant approach, no code is publicly available and computing these Shapley values requires significant computational time. Another relevant example is LimeOut (Bhargava et al., 2020) which uses LIME explanations to determine which features to drop to make a classifier fairer. This is a local and not global method, however, and the focus is on selecting features, not directly interpreting them via a feature importance score. In this paper, we are motivated to address these issues by proposing a very simple, intuitive, fast, and easy-to-compute fair feature importance score.

### 1.2 CONTRIBUTIONS

We make three major contributions that allow us to interpret a tree or tree-based model in terms of the fairness of its features. First, we propose and develop the first fair feature importance score (*Fair-FIS*) for interpreting decision trees. Second, we outline how to use *FairFIS* to interpret tree-based ensembles and tree-based global surrogates of complex ML systems. Finally, we empirically validate *FairFIS* for interpreting trees, tree-based ensembles, and tree-based surrogates of deep learning models on both synthetic and benchmark datasets.

## 2 *FairFIS*: FAIR FEATURE IMPORTANCE SCORE FOR TREES

### 2.1 REVIEW: FEATURE IMPORTANCE SCORE (FIS) FOR TREES

One of the many benefits of decision trees is that they have a straightforward mechanism for interpretation. The Feature Importance Score (*FIS*) is based on the Mean Decrease in Impurity (MDI), measured by a decrease in variance in regression or in the Gini Index or other metrics for classification Breiman (1973). Let us first introduce some notation to formally define and review *FIS*; this definition will help us in defining our Fair *FIS* in the next section. Suppose we have a response $y$ and the decision tree is built from data $\mathbf{X}$ based on $n$ samples. Additionally, let $t = 0$ be the root node

of the tree and $T$ be the total number of nodes in the tree; let $n_t$ be the number of samples falling in node $t$. Next, let $c_\ell(t)$ be the left child of $t$ and $c_r(t)$ be the right child of node $t$. Let $S_t = \{i \in t\}$ be the set of samples belonging to node $t$; let $y_{S_t}$ be the response associated with those samples in node $t$, we we denote as $y_t$ for ease of notation. Let $\hat{y}_t$ denote the predictions for samples in node $t$. As an example, for binary classification with $y \in \{0, 1\}$ or for regression with $\mathbf{y} \in \mathbb{R}$ recall that $\hat{y}_t = \frac{1}{|S_t|} \sum_{i \in S_t} y_i$; that is, $\hat{y}_t$ is the proportion of successes in node $t$ in the classification setting and the mean of node $t$ in the regression setting. Additionally, let $w_t$ represent the weighted number of samples $\frac{n_t}{n}$ at node $t$ and $\mathbb{1}_{\{(t,j)\}}$ denote the indicator that feature $j$ was split upon in node $t$. Let $\mathcal{L}(y, \hat{y})$ be the loss function employed to built the decision tree (e.g. MSE loss for regression or the Gini Index or Cross Entropy for classification). Now, we can formally define *FIS*:

**Definition 1.** *For a decision tree, the* FIS *(MDI) for feature $j$ is defined as:*

$$FIS_j = \sum_{t=0}^{T-1} \mathbb{1}_{\{(t,j)\}} (w_t \mathcal{L}(y_t, \hat{y}_t) - (w_{c_\ell(t)} \mathcal{L}(y_{c_\ell(t)}, \hat{y}_{c_\ell(t)}) + w_{c_r(t)} \mathcal{L}(y_{c_r(t))}, \hat{y}_{c_r(t)}))) \quad (1)$$

If feature $j$ is used to split node $t$, then the *FIS* calculates the change in the loss function before and after the split, or more precisely, the change in the loss between the predictions at node $t$ and the predictions of node $t$'s children. Hence, *FIS* uses the accuracy of the predictions to determine feature importance.

## 2.2 FAIRFIS

Inspired by *FIS*, we seek to define a feature importance score for group fairness that is based upon the bias of the predictions instead of the accuracy of the predictions. To do this, we first need to define group bias measures. Let $z_i \in \{0, 1\}$ for $i = 1, \ldots n$ be an indicator of the protected attribute (e.g. gender, race or etc.) for each observation. We propose to work with two popular metrics to measure the group bias, Demographic Parity (DP) and Equality of Opportunity (EQOP), although we note that our framework is conducive to other group metrics as well. In brief, DP measures whether the predictions are different conditional on the protected attribute whereas EQOP is typically only defined for classification tasks and measures whether the predictions are different conditioned on a positive outcome and the protected attribute (Hardt et al., 2016; Beutel et al., 2017).

One might consider simply replacing the loss function in equation 1 with these bias metrics, but constructing our fair metric is not that simple. Consider that for *FIS*, we can calculate the loss between $y_t$ and $\hat{y}_t$ for a particular node $t$, hence we can calculate the difference in loss after a split. We cannot use this same process, however, for bias as the predictions in each node of the decision tree are the same by construction. Thus, for a given node $t$, there are never any differences between the predictions based on protected group status. Hence, the bias calculated at node $t$ must always be zero. To remedy this and keep the same spirit as *FIS*, we propose to consider the difference in bias between the split that produced node $t$ and the split at node $t$ that produces node $t$'s children. Thus, we propose to calculate the bias that results from each split of the tree. To formalize this, notice that the result of each split in a tree is a right and left node. We call this set the level of the tree for node $t$ and denote this as $lev(t)$; this level includes the right and left node denoted as $lev_\ell(t)$ and $lev_r(t)$ respectively. We also let $c(t)$ denote all the children of $t$, or in other words, the child level of node $t$. Now, we can define our bias metrics for the split that produced node $t$, or in other words, the level of node $t$. The *Bias* of $lev(t)$ in terms of DP and EQOP are defined as follows:

$$Bias^{DP}(lev(t)) = \left| E(\hat{y}_i | z_i = 1, i \in lev(t)) - E(\hat{y}_i | z_i = 0, i \in lev(t)) \right|, \quad (2)$$

$$Bias^{EQOP}(lev(t)) = \big| E(\hat{y}_i = 1 | y_i = 1, z_i = 1, i \in lev(t))$$
$$- E(\hat{y}_i = 1 | y_i = 1, z_i = 0, i \in lev(t)) \big|. \quad (3)$$

These group bias metrics range between zero and one, with higher values indicating larger amounts of bias in the predictions. Armed with these definitions, we now seek to replace the loss function $\mathcal{L}$ in *FIS* with this *Bias* metric to obtain our *FairFIS*. To do so, we calculate the difference in bias between the level of node $t$ and node $t$'s children:

**Definition 2.** *The $FairFIS$ for feature $j$ is defined as:*

$$FairFIS_j = \sum_{t=0}^{T-1} \mathbb{1}_{\{(t,j)\}} w_t \left( Bias(lev(t)) - Bias(c(t)) \right) \quad (4)$$

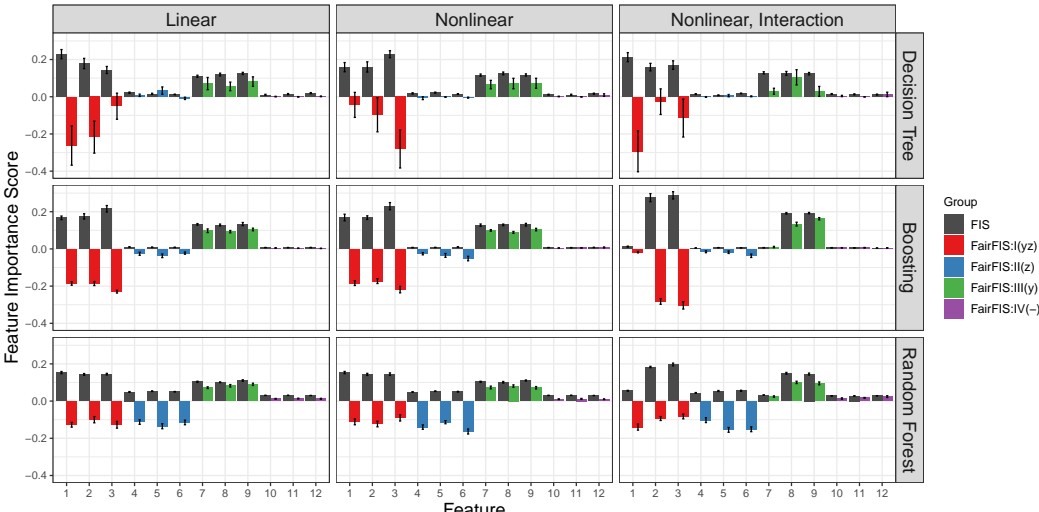

Figure 2: Classification results for *FIS* (MDI) using the Gini Index) and *FairFIS* (DP) on the three major simulation types and for a decision tree, gradient boosting, and random forest classifier. The magnitudes and directions of the *FairFIS* scores for each group align with what we would expect from the simulation construction, thus validating our metric.

Note that at the root node, $t = 0$, the level of the tree consists of only the root; then, $Bias(lev(0)) = 0$ for this constant model at the root in our definition. Finally, as the scale of *FIS* is not always interpretable, it is common to normalize *FIS* so that is sums to one across all features. We analogously do so for *FairFIS* by rescaling so that the sum of the absolute values across all features is one; in this manner, *FairFIS* and *FIS* are on the same scale and can be directly interpreted.

Our *FairFIS* formulation is an analogous extension of *FIS* as it calculates the *Bias* of the parent minus the *Bias* of the children summed over all splits that split upon feature $j$. But unlike *FIS* which is always positive, *FairFIS* can be both positive and negative. As decision trees are constructed with each split minimizing the loss, the difference in loss between parent and children is always positive. The splits do not consider *Bias*, however, so the *Bias* of the parent level could be higher or lower than that of the child level. Thus, *FairFIS* will be positive when the split at node $t$ improved the bias and negative when the split at node $t$ made the bias worse. *FairFIS* is then positive for features that improve the fairness (or decrease the bias) and negative for features that are less fair (or increased the bias). This is a particularly advantageous aspect of *FairFIS* that improves the interpretability of each feature with respect to fairness.

Figure 1 illustrates our *FairFIS* definition for a binary classification example. Panel A highlights our notation and calculation of *Bias* for levels of the tree. In Panel B, the bias improves from the parent level to the child level and hence *FairFIS* is positive, indicating the split improved the fairness of the predictions. The opposite happens in Panel C where the bias is worse in the child level and hence *FairFIS* is negative indicating worsening fairness as a result of the split.

In regression settings, *FairFIS* can be easily applied with the demographic parity metric equation 2, which is most commonly used for regression tasks. The bias can be calculated directly as the empirical mean of the predictions in each sensitive group. For classification settings, however, more care needs to taken in computing the *Bias* and our *FairFIS* metric, as discussed in the next section.

## 2.3  FAIRFIS IN CLASSIFICATION SETTINGS

Typically for classification tasks, people use hard label predictions to compute the DP and EQOP *Bias* metrics. However, for decision trees, this presents a problem as both the left and right node of level $t$ could predict the same hard label; the parent and child levels could also predict the same hard label. In these settings, using hard labels with equation 2 and equation 3 would result in zero or a misleading *Bias* measure even when the split might be unfair. This phenomenon is illustrated

in Figure 1 Panel D. To remedy this issue, we are left with two options: employ measures of *Bias* that take soft predictions or employ probabilistic decision trees that return stochastic hard label predictions based on the soft label probabilities. So that our *Bias* metrics are interpretable and comparable with others that typically employ hard label predictions, we choose the latter option. Let $lev_\ell(t)$ and $lev_r(t)$ denote the left and right nodes of the level of node $t$ and let $\pi_{lev_\ell(t)}$ and $\pi_{lev_r(t)}$ denote the proportion of positive samples in these nodes, respectively. Then for probabilistic trees, $\hat{y}_i$ for $i \in lev\ell(t)$ is a Bernoulli random variable with probability of success $\pi_{lev_\ell(t)}$, and that of the right node is defined analogously. Given this, we can directly apply equation 2 and equation 3 to compute the expectation necessary for our *Bias* metrics:

**Proposition 1.** *Consider binary classification with probabilistic trees, then our* Bias *measures are given by the following:*

$$
Bias^{DP}(lev(t)) = \left| \pi_{lev_\ell(t)} \left( \frac{\sum_i \mathbb{1}_{\{z_i=1, i \in lev_\ell(t)\}}}{\sum_i \mathbb{1}_{\{z_i=1, i \in lev(t)\}}} - \frac{\sum_i \mathbb{1}_{\{z_i=0, i \in lev_\ell(t)\}}}{\sum_i \mathbb{1}_{\{z_i=0, i \in lev(t)\}}} \right) \right.
$$
$$
\left. + \pi_{lev_r(t)} \left( \frac{\sum_i \mathbb{1}_{\{z_i=1, i \in lev_r(t)\}}}{\sum_i \mathbb{1}_{\{z_i=1, i \in lev(t)\}}} - \frac{\sum_i \mathbb{1}_{\{z_i=0, i \in lev_r(t)\}}}{\sum_i \mathbb{1}_{\{z_i=0, i \in lev(t)\}}} \right) \right|, \tag{5}
$$

$$
Bias^{EQOP}(lev(t)) = \left| \pi_{lev_\ell(t)} \left( \frac{\sum_i \mathbb{1}_{\{z_i=1, y_i=1, i \in lev_\ell(t)\}}}{\sum_i \mathbb{1}_{\{z_i=1, y_i=1, i \in lev(t)\}}} - \frac{\sum_i \mathbb{1}_{\{z_i=0, y_i=1, i \in lev_\ell(t)\}}}{\sum_i \mathbb{1}_{\{z_i=0, y_i=1, i \in lev(t)\}}} \right) \right.
$$
$$
\left. + \pi_{lev_r(t)} \left( \frac{\sum_i \mathbb{1}_{\{z_i=1, y_i=1, i \in lev_r(t)\}}}{\sum_i \mathbb{1}_{\{z_i=1, y_i=1, i \in lev(t)\}}} - \frac{\sum_i \mathbb{1}_{\{z_i=0, y_i=1, i \in lev_r(t)\}}}{\sum_i \mathbb{1}_{\{z_i=0, y_i=1, i \in lev(t)\}}} \right) \right|. \tag{6}
$$

Thus, even when employing probabilistic trees, our *Bias* measures and hence *FairFIS* is easy to compute. The proof / calculation for Proposition 1 is in the Supplemental materials. Note also that these results for the *Bias* and also *FairFIS* can easily be extended to multi-class classification settings, which we present in the Supplemental materials.

## 2.4 FAIRFIS FOR TREE-BASED ENSEMBLES AND DECISION TREE GLOBAL SURROGATES

Decision Trees are widely used due to their ability to break down complex problems into simpler solutions, thus making them more interpretable (Loh, 2011). Further, they are commonly employed in various popular ensemble-based classifiers such as random forest, gradient boosting, XGBoost, and others. For these tree-based ensembles, *FIS* is averaged (or averaged with weights) over all the trees in the ensemble (Breiman, 1996). We propose to extend *FairFIS* in the exact same manner to interpret all tree-based ensembles.

Decision trees have also gained attention for their role in knowledge distillation to transfer knowledge from large, complex models to smaller models that are easier to deploy (Hinton et al., 2015; Buciluă et al., 2006). Here, decision trees are not fit to the original labels or outcomes, but instead to the complex model's predicted labels or outcomes. Recently, others have proposed to use decision trees in a similar manner for global interpretation surrogates (Blanco-Justicia & Domingo-Ferrer, 2019; Yang et al., 2018; Sagi & Rokach, 2021; Wan et al., 2020). Decision trees are often an ideal surrogate in this scenario as a fully grown tree can exactly reproduce the predictions of the complex, black-box model. Hence, if the predictions match precisely, we can be more confident in the feature interpretations that the decision tree surrogate produces. Here, we propose to employ *FairFIS* to interpret features in a decision tree surrogate in the exact same manner as that of *FIS*. In this way, *FairFIS* provides a simple, intuitive, and computationally efficient way to interpret any large, complex, and black-box ML system.

## 3 EMPIRICAL STUDIES

### 3.1 SIMULATION SETUP AND RESULTS

We design simulation studies to validate our proposed *FairFIS* metric; these simulations are an important test since there are not other comparable fair feature interpretation methods to which we can

compare our approach. We work with four groups of features: features in $G_1$ and $G_2$ are correlated with the protected attribute $z$ and are hence biased, features in $G_1$ and $G_3$ are signal features associated with the outcome $y$, and features in $G_4$ are purely noise. We simulate the protected attribute, $z_i$, as $\mathbf{z} \overset{i.i.d}{\sim} Bernoulli(\pi)$ and take $\pi = 0.2$. Then, the data is generated as $\mathbf{x}_{i,j} \overset{i.i.d}{\sim} N(\alpha_j * z_i, \mathbf{\Sigma})$ with $\alpha_j = 2$ if $j \in G_1$ or $G_2$ and $\alpha_j = 0$ if $j \in G_2$ or $G_4$. Hence, all features in $G_1$ and $G_2$ are strongly associated with $z$ and hence should be identified as biased features with a negative *FairFIS*. Then, we consider three major simulation scenarios for both classification and regression settings: a linear model where $f(x_i) = \beta_0 + \sum_{j=1}^{p} \beta_j x_{ij}$, a non-linear additive scenario where $f(x_i) = \beta_0 + \sum_{j=1}^{p} \beta_j sin(x_{ij})$, and finally a non-linear scenario with pairwise interactions where $f(x_i) = \beta_0 + \sum_{j=1}^{p} \beta_j x_{ij} + \sum_{l=1,k=1}^{p} \gamma_{lk} sin(x_{il} x_{ik})$ and with $\gamma_{lk} = 1$ for the first two features in each group and zero otherwise. We also let $\beta_j = 1$ for $j \in G_1$ or $G_3$ and $\beta_j = 0$ for $j \in G_2$ or $G_4$. For regression scenarios, we let $y_i = f(x_i) + \epsilon$ where $\epsilon \overset{i.i.d}{\sim} N(0, 1)$, and for classification scenarios, we employ a logisitic model with $y_i \overset{i.i.d}{\sim} Bernoulli(\sigma(f(x_i)))$, where $\sigma$ is the sigmoid function. We present our binary classification results for the DP metric with $N = 1000$, $p = 12$ features, and $\mathbf{\Sigma} = \mathbf{I}$ in Figure 2. Additional simulation results for both classification and regression tasks with $N = 500$ or $1000$, larger $p$, correlated features with $\mathbf{\Sigma} \neq \mathbf{I}$, and for the EQOP metric are presented in the Supplemental Materials.

Figure 2 presents the *FIS* and *FairFIS* metric for each of the twelve features colored according to their group status, and averaged over ten replicates. We present all three simulation scenarios for decision tree, gradient boosting, and random forest classifiers. First, notice that the sign of *FairFIS* is correct in all scenarios; that is, features in $G_1$ (red) and $G_2$ (blue) are biased and *FairFIS* accurately reflects this bias with a negative score while the features in $G_3$ (green) and $G_4$ (purple) exhibit no bias and *FairFIS* is positive. *FairFIS* also accurately captures the magnitude of each feature's contributions as the magnitude of *FairFIS* and *FIS* are comparable in all scenarios. Note here that *FairFIS* values are low for non-signal features in trees and gradient boosting, as non-signal features are likely not split upon and hence do not contribute to bias or fairness. Because random forests use random splits, however, non-signal features are split upon more often and we see that *FairFIS* accurately determines that features in $G_2$ are biased. Overall, these results (and the many additional simulations in the Supplement) strongly validate the use of *FairFIS* for interpreting features in trees and tree-based ensembles with respect to the bias or fairness that the feature induces in the predictions.

## 3.2 CASE STUDIES

To align our work with the existing fairness literature, we evaluate our method on five popular benchmark datasets. We examine: (i) the Adult Income dataset (Dua & Graff, 2017) containing 14 features and approximately 48,000 individuals with class labels stating whether their income is greater than $50,000 and Gender as the protected attribute; (ii) the COMPAS dataset (Larson et al., 2022), which contains 13 attributes of roughly 7,000 convicted criminals with class labels that state whether the individual will recidivate within two years of their most recent crime and we use Race as the protected attribute; (iii) the Law School dataset (Whiteman, 1998), which has 8 features and 22,121 law school applicants with class labels stating whether an individual will pass the Bar exam when finished with law school and Race as the protected attribute; (iv) the Communities and Crimes (C & C) dataset (Dua & Graff, 2017), which contains 96 features of 2,000 cities with a regression task of predicting the number of violent crimes per capita and Race encoded as the protected attribute; and (v) the German Credit dataset, which classifies people with good or bad credit risks based on 20 features and 1,000 observations and we use Gender as the protected attribute.

We begin by validating the use of *FairFIS* for interpreting tree-based global surrogates. To do this, in Figure 3, we compare *FIS* and *FairFIS* results on a gradient boosting classifier (where these scores were calculated by averaging over all tree ensemble members) to *FIS* and *FairFIS* results for a tree-based surrogate of the same gradient boosting classifier (where a fully grown decision tree was fit to the model's predictions). Generally, we see that the *FIS* and *FairFIS* scores between the top row (boosting) and bottom row (surrogate) are similar in magnitude and direction. Specifically looking at the Adult dataset, we see that "Married" is an important feature according to *FIS* but *FairFIS* indicates that it is a highly biased feature; these results are reflected in both the boosting model and the tree surrogate. While the scores for some of the less important features may vary slightly between

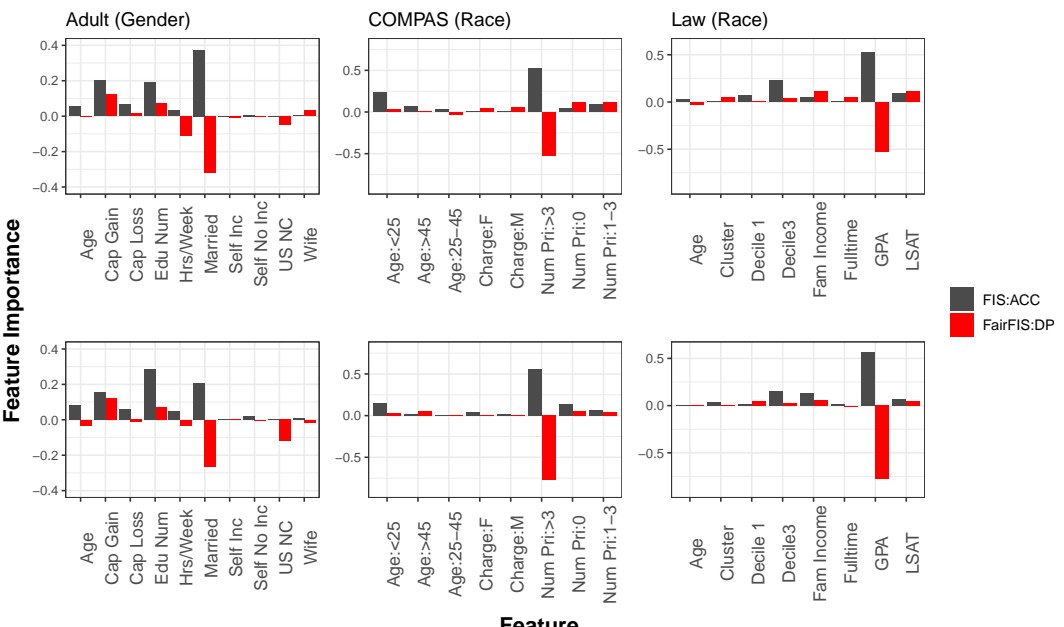

Figure 3: Global surrogate validation. The top row shows *FIS* and *FairFIS* results on a gradient boosting classifier for the Adult, COMPAS and Law datasets. The bottom row shows *FIS* and *FairFIS* results for a tree-based surrogate of a boosting classifier. The scores between the top and bottom rows are similar in magnitude and direction, indicating that our scores are effective when used to interpret tree-based global surrogates.

the original model and the surrogate, notice that the most important features are always consistent between the two approaches. This indicates that our *FairFIS* scores are effective when used to interpret tree-based global surrogates. Additional case studies on tree-based surrogates including validating *FIS* compared to model-specific deep learning feature importance scores are provided in the Supplemental Material.

Next, we evaluate the quality of *FairFIS* interpretations on several benchmark datasets in Figure 4; additional interpretations of all benchmarks are provided in the Supplemental Material. Panel A of Figure 4 shows scores for a tree-based surrogate of a deep learning model (multi-layer perceptron with two hidden layers each with $p$ units and ReLU activation) on the C & C dataset with Race as the protected attribute and the COMPAS dataset with Race as the protected attribute. In the C & C dataset, the percentage of kids who grew up with two parents in the household, denoted as "% Kids 2 Par", has the highest magnitude for both *FIS* and *FairFIS*, although *FairFIS* shows that this feature is strongly biased. Studies have shown that black young adults are disproportionately impacted by family structure (Wilcox, 2021). Specifically, black young adults are less likely to go to college and more likely to be imprisoned if they grow up in a single-parent household. In contrast, white young adults are significantly less affected by family structure. Thus, our *FairFIS* interpretations are consistent with these studies. Looking at the results for the COMPAS dataset, the number of priors greater than 3, denoted as "Num Pri > 3" has the highest magnitude for both *FIS* and *FairFIS*, and again *FairFIS* reveals that this feature is strongly biased. These interpretations are consistent with other studies on the COMPAS data set (Rudin et al., 2020), again validating our results.

In Panel B of Figure 4, we examine *FIS* and *FairFIS* scores for a tree-based surrogate of a deep learning model (multi-layer perception with two hidden layers each with $p$ units and ReLU activation) as well as a tree-based surrogate for a bias mitigation method, the Adversarial Debiasing approach (Zhang et al., 2018) for the Adult dataset with Gender as the protected attribute. The Adversarial Debiasing method (Zhang et al., 2018) applies adversarial learning to improve fairness by learning how to prevent an adversary from predicting the protected attribute. Looking at the Adult dataset scores of the tree-based surrogate of the deep learning model, we see that the "Cap. Gain", "Edu Num", and "Married" features are most important in terms of accuracy and US Native Country

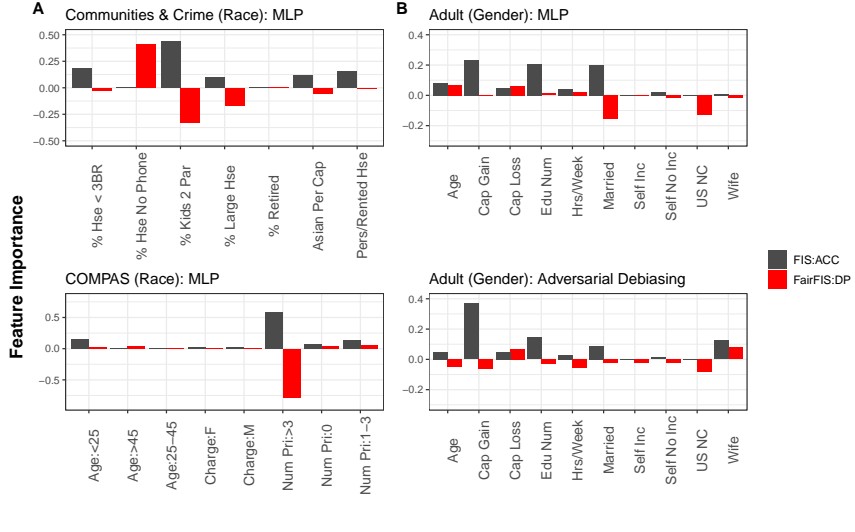

Figure 4: Interpretation of Features in Benchmark Datasets. Panel A displays the results of a tree-based surrogate of a deep learning model on the C & C dataset and the COMPAS dataset. Panel B explores the difference in importance scores between a tree-based surrogate of a deep learning model and a tree-based surrogate of a bias mitigation approach, Adversarial Debiasing.

("US NC"), "Married", and "Age" are most influential in terms of bias. Specifically, "US NC" and "Married" hurt the overall fairness of the model. In the debiasing method, the magnitude of both *FairFIS* and *FIS* for the feature "Married" decreases substantially, showing that using this feature likely would result in more biased predictions. Additionally, the "Cap. Gain" feature becomes more important in terms of accuracy in the debiasing model, as this feature exhibits relatively less bias. The accuracy and fairness go from 0.84 and 0.83 in the deep learning model to 0.80 and 0.92 in the Adversarial Debiasing model, indicating that the approach is successful at mitigating bias. Seeing as the severely unfair features become less unfair when the model becomes more fair indicates that our fair feature importance scores accurately capture when features are helping or hurting the overall fairness of the model. Note also that strongly predictive features often hurt fairness, and as fairness increases, accuracy decreases. This trend is a sign of the well-known and studied tradeoff between fairness and accuracy (Zliobaite, 2015; Little et al., 2022). Further results on all five benchmark datasets are included in the Supplemental material.

## 4 DISCUSSION

In this work, we proposed a fair feature importance score, *FairFIS*, for interpreting trees, tree-based ensembles, and tree-based surrogates of complex ML systems. We extend the traditional accuracy-based *FIS* (MDI), which calculates the change in loss between parent and child nodes, to consider fairness, where we calculate the difference in group bias between the parent and child levels. We empirically demonstrated that *FairFIS* accurately captures the importance of features with regard to fairness in various simulation and benchmark studies. Crucially, we showed that we can employ this method to interpret complex deep learning models when trees are used as surrogates.

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
