# OpenReview forum: "Fair Feature Importance Scores for Interpreting Tree-Based Methods and Surrogates"
_ICLR.cc/2024/Conference — Submitted to ICLR 2024_

### Official Review · Reviewer_oKhh · 2023-10-24

**Soundness:** 3 good
**Presentation:** 2 fair
**Contribution:** 2 fair
**Rating:** 5
**Confidence:** 4

**Summary:**

The paper proposes an approach for measuring the contribution of features to fairness metrics, namely equalized odds and demographic parity, in decision trees. This metric, called "Fair Feature Importance Score" (FairFIS), measures the change in group bias for a feature, akin to the traditional mean decrease in impurity. Through simulations and real-world tests on benchmark fairness datasets, the paper demonstrates the efficacy of FairFIS in providing valid interpretations for tree-based ensemble models and as surrogates for other ML systems.

**Strengths:**

The paper proposes a novel metric for quantifying the contributions to the overall feature measure based on the model features used in decision trees. Fairness is an important topic and the proposed metric uses equalized odds and demographic parity, which are two of the most commonly used metrics, making it appealing.

**Weaknesses:**

I think that the paper is fairly verbose and the presentation could be more concise. I had to go back and forth between pages to make sure that my understanding of the mathematical notation was correct. Nonetheless, the paper lacks two important discussions:
* How should practitioner use the proposed metric? A discussion is needed on this.
* Similarly to FIS, when features are correlated some may receive low FairFIS values even though they matter for fairness purposes.

In addition, I think that the method should be compared to other baselines. I can think of at least two:

* First, in all the datasets that the authors employed, the number of features is small. Thus, it’d be easy to fit p models where p is the number of features and each model is missing one feature, and then analyze how fairness metrics vary across models.  This idea is similar to leave-one-covariate-out (LOCO) inference, see https://www.stat.cmu.edu/~ryantibs/talks/loco-2018.pdf.
* Second, one could fit two different models to the data and compare the disparities of the models with the tree-based method proposed by https://arxiv.org/pdf/1707.00046.pdf. The authors could analyze FairFIS for each model and see if they reach similar conclusions about which features contribute towards disparities as they would with the method from that paper.

Other minor details:
* What are the error bars in Figure 2? Are they confidence intervals? What are the groups? The legend should be improved because it is mentioned that they are G1, G2 etc only on page 7.
* It seems that the authors are considering $\Sigma=I$ in the main paper, and $\Sigma\neq I$ in the Appendix. Is this correct?
* Proposition 1 could be shortened. It’s taking up a lot of space and it’s pretty clear how these metrics need to be computed.
* There are a couple of typos.

**Questions:**

Mentioned above.

---

> ### Author Response · Authors · 2023-11-15
>
> Thank you so much for your careful review and the positive comments on our work.
>
> - *How should practitioners use the proposed metric? A discussion is needed on this.*
>
> Thanks for the question and for giving us an opportunity to clarify this. FairFIS can be used in the same way that MDI (FIS) is used for decision trees and tree-based ensembles.  Also, FairFIS can be interpreted similarly to MDI (FIS) for decision trees, except that FairFIS can be both positive and negative.  A positive FairFIS score indicates that there are other biased features in the model and that this positively scored feature can help in reducing the impact from other biased features. On the other hand, features with negative FairFIS scores directly contribute to the bias of the model.  If a feature is not used in the model (e.g. not split upon in the tree), the FairFIS will be zero just as with FIS.  Thus, the sign of FairFIS indicates whether a feature improves or worsens model bias and the magnitude of FairFIS indicates the relative importance of the feature's contribution to the model fairness or bias.  In our paper, we use our FairFIS metric to interpret features in several case studies in Section 3.2.  We look forward to adding this discussion to our manuscript.
>
> - *Similarly to FIS, when features are correlated some may receive low FairFIS values even though they matter for fairness purposes.*
>
> Thank you for mentioning this important point. We completely agree that the MDI (FIS) for decision trees is known to have this problem; we also would like to note that this problem is somewhat mitigated in tree-based ensembles [1,2]. Further, dealing with correlated features is a known problem that is inherited by many widely used feature importance scores for accuracy (e.g. Shapley values, feature permutations, feature occlusion, and etc.).  Finally, we included a simulation with correlated features in the supplementary materials.  We found that our Fair FIS score performed as expected for tree-base ensembles with correlated features.
>
> -  *First, in all the datasets that the authors employed, the number of features is small. Thus, it’d be easy to fit p models where p is the number of features and each model is missing one feature, and then analyze how fairness metrics vary across models. This idea is similar to leave-one-covariate-out (LOCO) inference, see https://www.stat.cmu.edu/~ryantibs/talks/loco-2018.pdf.*
>
> Thank you for this comment and great suggestion. First, the datasets we study in our paper are standard benchmarks in fairness; these include many one-hot-encoded features which often measure in the hundreds and hence the number of features is not as small as it appears.  Next, LOCO inference or the related feature occlusion importance score are indeed very interesting approaches. To the best of our knowledge, however, these have not been proposed or studied in the context of fairness.  Feature occlusion importance requires refitting many models and is hence computationally expensive; our approach, on the other hand, is simple and fast.  We look forward to investigating LOCO / feature occlusion in the context of fairness in the future and to comparing it with our FairFIS metric. We will also add a discussion about this to our related works.
>
> - *Second, one could fit two different models to the data and compare the disparities of the models with the tree-based method proposed by https://arxiv.org/pdf/1707.00046.pdf. The authors could analyze FairFIS for each model and see if they reach similar conclusions about which features contribute towards disparities as they would with the method from that paper.*
>
> Thank you so much for this suggestion and reference.  This is an interesting method and as you suggest, we can use to compare two models with and without biased features and to further validate our metric and surrogates. We would like to highlight, however, the many simulation studies (many further studies are in the appendix) as well as case studies that we include in the paper to validate our approach.  Nonetheless, we look forward to investigating this model comparison framework in future work and will add a discussion about this method to our paper.
>
> Minor comments-
> - The error bars are the standard error.  We will also improve the clarity of the figure legend.
> - In the simulations presented in the main paper, the features are uncorrelated.  We include simulations with correlated features in the Appendix.
> -  Thanks for the suggestion regarding Proposition 1
> - Thanks for mentioning the typos and we look forward to fixing these.
>
> **Refereces**\
> [1] Strobl, Carolin, et al. (2007) *Bias in random forest variable importance measures: Illustrations, sources and a solution.* BMC bioinformatics 8.1: 1-21
>
> [2] Tianqi Chen, Micha ̈el Benesty, Tong He. 2018. *Understand Your Dataset with Xgboost.* https://cran.r-project.org/web/
> packages/xgboost/vignettes/discoverYourData.html#numeric-v.s.-categorical-variables

---

> > ### Comment · Reviewer_oKhh · 2023-11-21
> >
> > Thank you for your detailed explanations!
> >
> > By asking how practitioners should use this metric, I meant to ask what is the end goal of interpreting the features according to their FairFIS score. Is it bias mitigation, i.e., removing the most problematic features? If so, have the authors tried to do this and see if the model actually becomes more fair (in some sense) after the feature has been removed?
> >
> > I believe that the benchmarking exercise in the paper should at least compare the proposed approach with existing methods. While I agree with the authors that LOCO may be out of scope, the comparison with the tree-based method I pointed out should be conducted. As another reviewer points out, other explainability techniques can be applied in this context. Thus, some of them should be considered in the experiments.

---

> > > ### Author Response · Authors · 2023-11-22
> > >
> > > *By asking how practitioners should use this metric, I meant to ask what is the end goal of interpreting the features according to their FairFIS score. Is it bias mitigation, i.e., removing the most problematic features? If so, have the authors tried to do this and see if the model actually becomes more fair (in some sense) after the feature has been removed?*
> > >
> > > Thank you for your question regarding using FairFIS in practice. A practitioner should use this proposed metric to understand how features contribute to the fairness or bias in a model's predictions.  Interpreting features with respect to fairness is useful for promoting societal trust via transparency and interpretability in an ML system, auditing an ML system for potential bias, and for model development and debugging to improve model fairness.  Selecting features is related to the latter, and our FairFIS metric can certianly be used to idetify and potentially remove extremely biased features.  Thanks for the suggestion for the additional empirical study to see if the fairness imrpoves after removing the most biased features identified via FairFIS; we look forward to adding this comparison to our paper.

---

### Official Review · Reviewer_rM4n · 2023-10-30

**Soundness:** 2 fair
**Presentation:** 3 good
**Contribution:** 2 fair
**Rating:** 5
**Confidence:** 2

**Summary:**

In this work, the authors introduced the limited prior work on the understanding of how a feature affects the fairness of the model's predictions. The authors then proposed FairFIS, a surrogate model to interpret tree-based model which assigned the fairness score to each of the features, providing the protected atrribute are provided. The authors demonstrated in simulation and real data that FairFIS is capable of capturing the important features that increase/ decrease the bias w.r.t the protected variables in their definition.

**Strengths:**

The question the authors attempted to answer is novel and important: Given a protected feature, what are the contribution of other features with respect to minimizing  the contribution of the protected feature. This is a great approach to understanding the fluctuation of feature contribution w.r.t. a particular feature that is not of interests.

**Weaknesses:**

1. The paper's approach to addressing fairness through the proposed algorithm is not adequately substantiated. The algorithm requires users to define a set of "protected" features, then computes bias based on this selection. This methodology rests on the assumption that the chosen protected features are inherently unbiased—a claim not demonstrated in the paper. To bolster the argument that the FairFIS algorithm genuinely addresses fairness, the authors should either provide a rigorous formal definition of what constitutes a "protected feature" or temper their claims regarding the algorithm's ability to ensure fairness.

2. The generative model used in the simulation doesn't align with the authors' objective. In the generative model the authors proposed, they first defined the protected attribute z, and then generate x from z, and finally generate y from x. The graph can be expressed as z --> x --> y. This implicitly suggested that z is the root cause of y. Given this assumption, it is not clear why "z" should be designated as a "protected" attribute in the first place. Otherwise please simply substitute z as a variable should be protected (i.e. race) and y as any sensitive subject and see what happens.

**Questions:**

How should one select the protected features? Who should decide the fairness of the selection of protected features? Are income, country, religion considered as protected features?

---

> ### Author Response · Authors · 2023-11-15
>
> Thank you for your review.
>
> - *The paper's approach to addressing fairness through the proposed algorithm is not adequately substantiated. The algorithm requires users to define a set of "protected" features and then computes bias based on this selection. This methodology rests on the assumption that the chosen protected features are inherently unbiased—a claim not demonstrated in the paper. To bolster the argument that the FairFIS algorithm genuinely addresses fairness, the authors should either provide a rigorous formal definition of what constitutes a "protected feature" or temper their claims regarding the algorithm's ability to ensure fairness.*
>
> Thank you for the question. We would like to clarify that in terms of fairness, protected attributes are those traits that are prohibited by law from being the subject of discrimination. These include gender, race, age, and etc.  These protected attributes are prohibited from being used for decision-making and hence cannot be used in the training of any machine learning model.  We look forward to clarifying this in our paper.
>
>
> - *The generative model used in the simulation doesn't align with the authors' objective. In the generative model, the authors proposed, they first defined the protected attribute z, and then generate x from z, and finally generate y from x. The graph can be expressed as z --> x --> y. This implicitly suggested that z is the root cause of y. Given this assumption, it is not clear why "z" should be designated as a "protected" attribute in the first place. Otherwise please simply substitute z as a variable should be protected (i.e. race) and y as any sensitive subject and see what happens.*
>
> Thank you for the question. In the generative model, z can be treated as any binary protected attribute (e.g. male or female). We have four groups of features that constitute x. The specific relation that you mentioned is true for group 1. The features of Groups 1 and 2 are correlated to the protected attribute z and so they should have a negative FairFIS score. The features that are not correlated to z (Groups 3 and 4), should have positive FairFIS scores. Our simulation results show these trends. Thank you for giving us a chance to further clarify the simulations.

---

> > ### Comment · Reviewer_rM4n · 2023-11-21
> >
> > Thanks for the authors' explanation.
> >
> > "We would like to clarify that in terms of fairness, protected attributes are those traits that are prohibited by law from being the subject of discrimination. These include gender, race, age, and etc."
> >
> > The authors should have spent a considerable amount of effort discussing this. Firstly, it is unclear whether an attribute should be protected or not without a concrete response variable. It is unclear whether gender, race, and age are important factors in the occurrence of complex diseases or should be protected instead. Second, how many established attribute-response pairs are there that should be protected, and should they be protected under all contexts? Without offering an explicit answer to this question, the actual application of FairFIS can be limited.
> >
> > "The features of Groups 1 and 2 are correlated to the protected attribute z and so they should have a negative FairFIS score."
> >
> > Let me ask the question in another way: say in another simulation setting, we define z as a protected attribute. y will be entirely dependent on features in Group 1 and Group 2 (i.e., y is not dependent on features from Groups 3 and 4). In this case, does FairFIS give all the contributing features negative scores? If so, how should we interpret the result?

---

### Official Review · Reviewer_q1kB · 2023-10-31

**Soundness:** 3 good
**Presentation:** 2 fair
**Contribution:** 2 fair
**Rating:** 5
**Confidence:** 3

**Summary:**

In this paper, the authors proposed a new feature importance score to investigate whether bias exists in the machine learning models. This score is adapted from the classical feature importance score over tree-based models by considering the difference between the bias of the nodes and their children. Through some experiments in the simulated settings, the authors can demonstrate that the proposed metric can provide reasonable explanations of why bias occurs in the predictions of the machine learning models.

**Strengths:**

+ The problem of interpreting why and how bias occurs in machine learning systems is an important problem to study
+ The authors proposed a simple and generic solution to solve this problem
+ The authors tried to provide extensive experiments to empirically demonstrate the benefits of the proposed solution.

**Weaknesses:**

+ I have some concerns about the motivations of this paper. The authors claim that they want to interpret how a feature influences the prediction bias of one model. However, I feel that this problem could be solved by performing counterfactuals over the features that we want to explain and evaluating how the prediction bias gets changed. The authors need to provide more justifications for why this strategy is not satisfactory.
+ Related to the above point, the baseline comparison in the experiments is very simple and lacks many critical baseline methods, in particular, the state-of-the-art feature importance score over general models. As mentioned above, we adapt such baseline methods by measuring how perturbing the target feature impacts the prediction bias. It would be essential to consider the adaptations of such a baseline and perform an empirical comparison
+ Interpreting the influence of a feature over the model prediction bias through the decision tree surrogate is also problematic. I am not sure whether the feature that causes the most significant prediction bias for the surrogate model can really cause the same amount of bias in the original model. The authors need to verify this somehow, say through performing counterfactuals over the target features on the original models.
+ I am also worried about the novelty of the proposed solution. In my mind, it is still like simply replacing the loss function in FIS score with bias metrics, which seems to be straightforward to me. More discussion on why the proposed metric is non-trivial is needed.

**Questions:**

See above.

---

> ### Author Response · Authors · 2023-11-15
>
> Thank you for your comments regarding the importance of this problem and the simple nature of our proposed solution.
>
> -  *I have some concerns about the motivations of this paper. The authors claim that they want to interpret how a feature influences the prediction bias of one model. However, I feel that this problem could be solved by performing counterfactuals over the features that we want to explain and evaluating how the prediction bias gets changed. The authors need to provide more justifications for why this strategy is not satisfactory.*
>
> Counterfactual explanations are a really interesting approach to understand a feature's contribution to fairness; thanks for this suggestion.  To the best of our knowledge, these have not been explored before in the context of fairness.  In this paper, our goal was to develop a global feature importance score.  Notice that counterfactual explanations are local in nature and hence cannot provide a global metric for fair feature importance. As a result, counterfactual explanations could be more appropriate for individual fairness (local) rather than group fairness (global) which we study in this paper. We look forward to adding a discussion of this to our related works section.
>
>
> - *Related to the above point, the baseline comparison in the experiments is very simple and lacks many critical baseline methods, in particular, the state-of-the-art feature importance score over general models. As mentioned above, we adapt such baseline methods by measuring how perturbing the target feature impacts the prediction bias. It would be essential to consider the adaptations of such a baseline and perform an empirical comparison*
>
> From what you are describing, it sounds like you are referring to feature permutation scores. If so, this is a great idea and has been studied for how feature permutation impacts accuracy. To the best of our knowledge, however, these have not yet been applied in the context of fairness. While there are many advantages, feature permutations for accuracy are known to have a few weaknesses including expensive computation, poor performance for correlated features, and they often create features that are outside of the domain of the original data.  Feature permutations for fairness would likely inherit these properties and hence we believe a thorough examination of these and their properties would be needed before we should use them as a baseline for comparison.  Our FairFIS approach, on the other hand, is simple, computationally fast, and inherits some of the nice properties of tree-based feature importance scores. The idea of feature permutation scores will be useful to include in our discussion and we look forward to exploring this further in future work.
>
> - *Interpreting the influence of a feature over the model prediction bias through the decision tree surrogate is also problematic. I am not sure whether the feature that causes the most significant prediction bias for the surrogate model can really cause the same amount of bias in the original model. The authors need to verify this somehow, say through performing counterfactuals over the target features on the original models.*
>
> Thank you for this comment. We would like to draw your attention to a figure in the appendix where we validate our surrogate model by comparing the accuracy-based feature importance scores from a tree-based surrogate of a deep learning model with scores from epsilon-Layerwise Relevance Propagation (LRP). In the future, we plan to investigate this further and include comparisons with other feature importance scores.
>
> - *I am also worried about the novelty of the proposed solution. In my mind, it is still like simply replacing the loss function in FIS score with bias metrics, which seems to be straightforward to me. More discussion on why the proposed metric is non-trivial is needed.*
>
> Thank you for your question regarding novelty. We consider our main novelty to be the idea of developing simple and effective scores as well as the implementation of using these scores with surrogates. Another contribution is that we also provide several case studies on real datasets to show how to effectively use FairFIS for interpretation.

---

### Official Review · Reviewer_Vvdg · 2023-11-10

**Soundness:** 1 poor
**Presentation:** 2 fair
**Contribution:** 1 poor
**Rating:** 3
**Confidence:** 4

**Summary:**

The paper focuses on the problem of explaining the unfairness of ML models. The proposed metric works for tree based models only. They key idea is to slightly alter the node splitting procedure to compute the unfairness score of a feature.

**Strengths:**

1. The paper correctly notes that using explanations to understand the unfairness of a model is an important desideratum in applications of ML.

2. The proposed procedure is simple and easy to understand.

**Weaknesses:**

While it focuses on an interesting and timely problem, I think the paper still has some key issues which should be addresses before its ready for publication.

1. **Framing:** The paper claims to be the first to consider fairness and explainability of ML models. For instance, the paper notes that "we have no current way of understanding how a feature affects the fairness of the model’s predictions". However, there is already non-negligibly amount of work that focuses on fairness and explainability both. Consider for example [this blogpost](Explaining Measures of Fairness) showing how to use SHAP to understand model unfairness. On a more academic side, I would suggest that the paper factors papers 1-4 (and related references therein) in the related work section so that the readers can better frame its contributions.

2. **Motivation:** The paper needs to motivate the design choices in a better way. I am not sure that "trees have a popular and easy-to-compute intrinsic feature importance score known as mean decrease in impurity" is the most important reason to focus on trees. There exist plenty of methods like SHAP, LIME and Integrated Gradients for explaining all kinds of other ML models. Similarly, why consider the mean decrease in impurity (MDI) score? There are other methods like TreeSHAP which seem to offer better theoretical properties. Also, the proposed FairFIS metric has a rather interesting choice in comparing a parent and the child node. The paper explains why the the computation of the style of Eq. 1 was not considered, but does not mention what are the pros and cons of the computation of FairFIS? How can we ascertain that this choice corresponds to how humans would expect model explanations to behave?

3. **Evaluation:** The evaluation, while considering multiple datasets, is quite high level and relies on the parallels between FIS and FairFIS. I would suggest performing a more systematic analysis and consider quantitative evaluations metrics (e.g., those considered [here](https://arxiv.org/abs/1705.07874) and [here](https://arxiv.org/abs/1912.09405)).

4. **Writing:** I think the writing should also be improved before the paper is ready for publication. Currently, the paper tends to simply provide information without first giving motivation and reasoning. Consider for instance the experiment in Section 3.1, where the paper directly dives into the details of the experiment without explaining why it was set up in this way, what metrics should the users watch out for, and what kind of values of should they expect to see.

[1] [What will it take to generate fairness-preserving explanations?](https://arxiv.org/pdf/2106.13346.pdf)

[2] [Biased Models Have Biased Explanations](https://arxiv.org/pdf/2012.10986.pdf)

[3] [Cohort Shapley value for algorithmic fairness](https://arxiv.org/pdf/2105.07168.pdf)

[4] [Exploring the usefulness of explainable machine learning in assessing fairness](https://theses.ubn.ru.nl/server/api/core/bitstreams/b019c49d-b4e9-4f81-a5e6-ac4fadcdbf0f/content)

**Questions:**

Please see points 1-3 in the "Weaknesses" section.

---

> ### Author Response · Authors · 2023-11-15
>
> Thank you for acknowledging the importance of this problem and noting the clarity of our proposed metric.
>
> 1) **Framing**
> *The paper claims to be the first to consider fairness and explainability of ML models. For instance, the paper notes that "we have no current way of understanding how a feature affects the fairness of the model’s predictions". However, there is already non-negligibly amount of work that focuses on fairness and explainability both. Consider for example [this blogpost](Explaining Measures of Fairness) showing how to use SHAP to understand model unfairness. On a more academic side, I would suggest that the paper factors papers 1-4 (and related references therein) in the related work section so that the readers can better frame its contributions.*
>
> Thank you for bringing these interesting papers to our attention.  While this literature tells us important information about the features, they do not provide a global score or metric. Furthermore, the methods inspired by SHAP tend to be very computationally expensive, especially with large numbers of features. On the other hand, FairFIS is fast, easy to compute, and provides a global metric for fair feature interpretation. We look forward to adding further discussion of these papers to our related works and contextualizing them with our proposed method.
>
> 2. **Motivation** *The paper needs to motivate the design choices in a better way. I am not sure that "trees have a popular and easy-to-compute intrinsic feature importance score known as mean decrease in impurity" is the most important reason to focus on trees. There exist plenty of methods like SHAP, LIME and Integrated Gradients for explaining all kinds of other ML models. Similarly, why consider the mean decrease in impurity (MDI) score? There are other methods like TreeSHAP which seem to offer better theoretical properties. Also, the proposed FairFIS metric has a rather interesting choice in comparing a parent and the child node. The paper explains why the the computation of the style of Eq. 1 was not considered, but does not mention what are the pros and cons of the computation of FairFIS? How can we ascertain that this choice corresponds to how humans would expect model explanations to behave?*
>
> Our motivation was to develop a simple, fast, easy-to-use, and easy-to-interpret metric for trees and tree-based ensembles.  Since the MDI score for trees is the most widely used and the fastest, we opted to focus on this.  While SHAP, TreeSHAP, LIME and many other model-specific methods can be useful, they are typically computationally expensive, which is why we chose to employ the MDI approach for trees. Computationally, FairFIS has many benefits. Just as in FIS, we can automatically calculate the change in bias for each split of the tree and sum over all of the splits that contain feature $j$. This is fast computationally and a major motivation for our work.
>
> 3. **Evaluation** *The evaluation, while considering multiple datasets, is quite high level and relies on the parallels between FIS and FairFIS. I would suggest performing a more systematic analysis and consider quantitative evaluations metrics (e.g., those considered here and here).*
>
> Thank you for the suggestions and references. We chose to compare mainly to FIS since our method is derived from and inspired by FIS. Because there are so few baselines, we decided to build our own generative model to test our approach. Our simulation model has four feature groups to better understand the behavior of FairFIS for different types of features. Specifically, group 1 is correlated with both the protected attribute and the outcome, group 2 is correlated only with the protected attribute, group 3 is correlated with the outcome, and group 4 is noise. As reflected in our simulation results, the FairFIS for groups 1 and 3 are negative because those features are correlated with the protected attribute. In the future, we look forward to adding more baseline comparisons.
>
> 4. **Writing** *I think the writing should also be improved before the paper is ready for publication. Currently, the paper tends to simply provide information without first giving motivation and reasoning. Consider for instance the experiment in Section 3.1, where the paper directly dives into the details of the experiment without explaining why it was set up in this way, what metrics should the users watch out for, and what kind of values of should they expect to see.*
>
> Thank you very much for this comment and we look forward to improving this aspect of the paper. Please see our point three for further discussion on how we set up this model.

---

### Meta-Review · Area_Chair_Z4hU · 2023-12-12

**Metareview:**

This paper looks at the intersection of XAI and fair ML. Specifically, it looks to use interpretable-by-design tree-based methods to cover some of the same ground that feature importance scores do.  From this AC's point of view, that's a nice idea -- but there are many critical gaps brought up by reviewers.  All reviewers agreed the paper's motivation, framework design, and experimental design need to be more exact and more grounded in both the literature and the reality of fair ML and XAI.  We encourage the authors to add to their strong basis to improve the work.

**Justification For Why Not Higher Score:**

Reviewers broadly found issue with the paper.  One reviewer had unanswered questions (after interaction) with the post-rebuttal process.

**Justification For Why Not Lower Score:**

N/A

---

### Decision · Program_Chairs · 2024-01-16

Reject